# *In Vitro* Susceptibility of *Cryptosporidium parvum* to Plant Antiparasitic Compounds

**DOI:** 10.3390/pathogens12010061

**Published:** 2022-12-30

**Authors:** Sandamalie Ranasinghe, Alireza Zahedi, Anthony Armson, Alan J. Lymbery, Amanda Ash

**Affiliations:** 1Centre for Biosecurity and One Health, Harry Butler Institute, Murdoch University, Perth 6150, Australia; 2Psychology, Counselling, Exercise Science and Chiropractic, College of Science, Health, Engineering and Education, Murdoch University, Perth 6150, Australia; 3Centre for Sustainable Aquatic Ecosystems, Harry Butler Institute, Murdoch University, Perth 6150, Australia

**Keywords:** Ayurveda, cytotoxicity, cryptosporidiosis, drug screening, plant medicine

## Abstract

*Cryptosporidium parvum* is a significant cause of watery diarrhoea in humans and other animals worldwide. Although hundreds of novel drugs have been evaluated, no effective specific chemotherapeutic intervention for *C. parvum* has been reported. There has been much recent interest in evaluating plant-derived products in the fight against gastrointestinal parasites, including *C. parvum*. This study aimed to identify extracts from 13 different plant species that provide evidence for inhibiting the growth of *C. parvum* in vitro. Efficacy against *C. parvum* was detected and quantified using quantitative PCR and immunofluorescence assays. All plant extracts tested against *C. parvum* showed varying inhibition activities in vitro, and none of them produced a cytotoxic effect on HCT-8 cells at concentrations up to 500 µg/mL. Four plant species with the strongest evidence of activity against *C. parvum* were *Curcuma longa*, *Piper nigrum, Embelia ribes*, and *Nigella sativa,* all with dose-dependent efficacy. To the authors' knowledge, this is the first time that these plant extracts have proven to be experimentally efficacious against *C. parvum*. These results support further exploration of these plants and their compounds as possible treatments for *Cryptosporidium* infections.

## 1. Introduction

Species of the enteric protozoan parasite *Cryptosporidium* are associated with morbidity and mortality in people, particularly in developing countries. Of the 46 species characterised, *Cryptosporidium parvum* and *Cryptosporidium hominis* cause the most human infections [1,2]. In immunocompetent adults, symptoms generally resolve within two weeks; however, the infection can be critical in immunocompromised patients and young children. Recent studies estimate that 30–50% of children under the age of five are infected with *Cryptosporidium* spp. worldwide [3], and it is the second most common cause of moderate-to-severe diarrhoea in this age group [4]. In addition, *Cryptosporidium* spp. are among the five pathogens with the highest attributable burden of diarrhoea in the first two years of life [5], with deaths estimated to be 60,000–200,000 annually [6].

*Cryptosporidium* infection affects many animal hosts worldwide, including livestock such as cattle, sheep, goats, and pigs [7]. Neonatal (<6 weeks old) calves are very vulnerable to cryptosporidiosis [8]. The infected animals suffer from a gastrointestinal disorder that is often accompanied by diarrhoea of variable duration. In livestock, the disease may lead to significant production losses due to retarded growth and mortality of the animals, the cost of drugs, veterinary assistance, and increased staff labour [9]. Zoonotic transmission is evident in many *Cryptosporidium* spp. [10,11]. *Cryptosporidium parvum* is the most important zoonotic species [12].

Although hundreds of novel drugs have been evaluated for anti-cryptosporidial activity, the current treatment options are limited, with nitazoxanide the only drug permitted by the US Food and Drug Administration (FDA) [13] for the treatment of cryptosporidiosis. However, this drug has exhibited minimal efficacy in immunocompetent adults and children and is ineffective in immunocompromised patients with acquired immunodeficiency syndrome (AIDS) [14].

Thus, the need for new treatment options against *Cryptosporidium* spp., which are safe, effective, and cheap, is imperative to deal with the current global burdens of cryptosporidiosis effectively. Natural products with antiparasitic activity could play an important role in combating these protozoan parasites. Traditional medicinal plants and their derivatives have shown favourable results against various parasitic infections [15,16,17].

In vitro screening is a critical step in the early stage of drug development. Compared to animal models, in vitro models are less expensive, less time-consuming, and more convenient for screening drugs for their efficacy against parasites [18,19]. Using in vitro cultures to assess the infectivity and proliferation of the parasite may also help reduce the use of experimental animal models [19,20]. Various cell lines have been successfully applied for in vitro studies on *C. parvum,* with human ileocaecal adenocarcinoma (HCT-8) cells having been proven to be a suitable cell line for developing the complete life cycle of *C. parvum* [21,22,23,24].

In searching for new antiparasitic agents, in the current study, we tested thirteen plant extracts for their ability to inhibit the growth of *C. parvum*, established in vitro on HCT-8 cells. The extracts were chosen because of their commercial availability, diversified pharmacological properties, and traditional use against various diseases, including parasitic gastrointestinal (GI) disorders.

## 2. Materials and Methods

### 2.1. Parasites

*Cryptosporidium parvum* oocysts were purchased from BioPoint Pty. Ltd., Belrose NSW, Australia (Batch No: 859–547) and stored in sterile-filtered phosphate-buffered saline (PBS) containing 100 µg of streptomycin/mL and 100 U penicillin/mL at 5 °C until used. Oocysts were <6 months old when used.

### 2.2. Plant Material

Thirteen plant extracts were used in this study (Table 1). These were selected based on ethnopharmacological information on their use as antiparasitic treatments in Traditional Indian Medicine (Ayurveda) [25,26]. The dried powder form of plant extracts of *Allium sativum* (garlic), *Cucurbita pepo* (pumpkin), *Embelia ribes* (false black pepper), and *Thymus vulgaris* (thyme) was commercially purchased from Flamingo Exports, India. The remaining nine dried plant extracts were supplied from Sunpure Extract Ltd., India. All the plant extracts were standardised and validated by the manufacturers using high-performance liquid chromatography (HPLC) and the principal phytochemicals in each extract are shown in Table 1. Dried plant extracts were stored in airtight dark containers at room temperature. The maximum dose range tested for all compounds was 500 µg/mL, as compounds could not be completely dissolved in absolute dimethyl sulfoxide (DMSO) beyond this concentration.

### 2.3. Cell Culture

Human ileocaecal adenocarcinoma cells, originally obtained from American Type Culture Collection (ATCC; CCL-244™), Manassas, Virginia, were used for the in vitro maintenance of *C. parvum* infections and cultured in 25 cm^2^ sterile polystyrene ventilated cell culture flasks (Cellstar^®^) in growth media consisting of RPMI 1640 (Sigma-Aldrich^®^) medium (10.3 g/L) supplemented with 10% foetal calf serum (FCS) (Sigma-Aldrich^®^) and other supplements as previously described by Hijjawi et al. [27]. The flasks were kept in a humidified incubator at 37 °C and 5% (*v*/*v*) CO_2_ and grown for 24 h until monolayers reached 90% confluence.

### 2.4. Cytotoxicity Assay

The cytotoxicity of all plant extracts was determined before evaluating their anti-cryptosporidial effects.

The cytotoxicity of plant compounds was evaluated on HCT-8 cells using Cell Titer-Blue^®^ cell viability assay (cat # G8081, Promega Corporation, Madison, WI, USA), following the manufacturer’s instructions. This assay offers a homogeneous, fluorometric method to estimate the number of viable cells. The Cell Titer-Blue^®^ (CTB) Reagent contains highly purified resazurin dye, which enables the measurement of cells’ metabolic capacity as an indicator of cell viability. Viable cells can reduce resazurin, which is dark blue in colour and non-fluorescent, into resorufin, which is pink and highly fluorescent. Nonviable cells cannot reduce the indicator dye as they lose metabolic capacity and cannot generate a fluorescent signal [28,29].

To determine cytotoxicity, HCT-8 cells were seeded at a density of 4 × 10^4^ cells/well in 100 µL medium per well and incubated overnight to grow at 37 °C and 5% CO_2_. Plant extracts were redissolved in 100% DMSO (ChemSupply Australia, Gillman, SA, Australia), and the concentration was adjusted to 500 µg/mL. The plates with 90% confluent cell monolayers were incubated at 37 °C and 5% CO_2_ in the presence of a three-fold dilution of extracts, ranging from 2–500 µg/mL. After a minimum of 66 h incubation, monolayers were checked for any signs of contamination and 20 µL of CTB was added to each well. The plates were shaken for 10 s and then incubated at 37 °C and 5% CO_2_ for 4 h. Once the colour change was observed, the plates were shaken for 10 s and read on the DTX 800 Multimode Detector (Beckman Coulter’s Biomek^®^) at 560 nm excitation/590 nm emission. All plant extracts, at all concentrations, were tested in three independent replicate experiments. In each replicate, cells not exposed to the compounds served as the untreated control, and the wells that contained only the media served as the background control to determine background fluorescence that may be present. Five µL DMSO was added to each untreated and background control well. Each experimental plate had four untreated controls and six background controls.

### 2.5. Cryptosporidium parvum Growth Inhibition Assay

#### 2.5.1. Pre-Treatment of Oocysts and Infection of Host Cells

The required number of *C. parvum* oocysts was bleached using 200 µL sodium hypochlorite (Scott Scientific, Western Australia) in 10 mL of water at room temperature for 30 min, centrifuged at 2000 g for 8–10 min, and the supernatant was removed. Bleached oocysts were then placed into 10 mL of freshly prepared, filter sterilised (0.22 μm filter), and warm (37 °C) excystation media composed of acidic water (pH 2.5–3) containing 0.5% trypsin/EDTA (Sigma-Aldrich^®^). The oocysts were incubated for 30 min at 37 °C with vigorous shaking every 5 min. The excystation suspension was then centrifuged (at 2000× *g* for 8–10 min) and resuspended in the appropriate volume of maintenance media composed of RPMI 1640 (Sigma-Aldrich^®^) medium, 1% FCS and other supplements as described previously [27].

Infectivity screens were set up by detaching HCT-8 cell monolayers from culture flasks using 1 mL of trypsin/EDTA (Sigma-Aldrich^®^) and inoculating into 48-well plates (CELLSTAR^®^) at a density of 10^4^ cells/well in 500 µL of RPMI growth medium. The plates were then incubated at 37 °C and 5% CO_2_ overnight to allow them to reach the monolayer stage. Following overnight incubation, the growth medium was removed from the monolayers by aspiration, prepared oocysts were applied to the monolayers (7500 oocysts/well), and the monolayers were incubated overnight at 37 °C and 5% CO_2_. Infectivity was confirmed using EasyStain™ (BioPoint Pty. Ltd., Belrose, NSW, Australia) (Appendix A). Two wells on each plate were left uninfected (i.e., oocysts were not added) to serve as negative controls in extract screening tests (see below).

#### 2.5.2. Extract Screening at a Single Concentration

Initially, the efficacy of all 13 plant extracts in inhibiting parasite growth was tested at a standard concentration of 500 ng/mL, with three independent replicate experiments for each extract. Screening plates were prepared as described in Section 2.5.1. Eight plant extracts were tested on one plate. Each experimental plate included two negative control wells (containing uninfected cell monolayers) and six positive control wells (infected but untreated cell monolayers). Five µL DMSO was added to each negative and positive control well. Compound stock solutions with 500 µg/mL initial concentration were diluted 1:10 (10 µL compound + 90 µL DMSO) making them 50 µg/mL. From this, 5 µL was added to the 500 µL RPMI growth media in the well, giving a further dilution of 1:100, making the final concentration 500 ng/mL. The plates were incubated at 37 °C and 5% CO_2_ for 48 h.

#### 2.5.3. Dose–Response Analysis

The four plant extracts showing the greatest anti-cryptosporidial activity (*C. parvum* inhibition >50%) at 500 ng/mL were further examined to produce a dose–response curve and consequently ascertain an IC_50_. Three independent replicate experiments were conducted for each extract. Screening plates were prepared as described in Section 2.5.1. A single plant extract was tested on one plate. A three-fold serial dilution (5 µL/well) was created in the screening plate, initiating from 500 ng/mL, giving extract concentrations of 166.7, 55.6, 18.5 and 6.1 ng/mL. Each experimental plate included two negative control wells (containing uninfected cell monolayers) and six positive control wells (infected but untreated cell monolayers). As before, 5 µL DMSO was added to the negative and positive controls. Trifluralin (Sigma-Aldrich^®^) was used as a standard drug control. Trifluralin is a dinitroaniline, a promising class of anti-cryptosporidial compounds with tubulin-binding properties that were initially recognised for their herbicidal properties [20,30]. Trifluralin has been shown to be effective in inhibiting *C. parvum* growth in vitro (IC_50_ = 650.4 mg/L) [30]. A 1 × 10^5^ ng/mL (in acetonitrile) stock solution of trifluralin was diluted 1:10 with DMSO), making it 1 × 10^4^ ng/mL and 5 µL was added to the 500 µL RPMI media in the well. Then, a three-fold serial dilution was created in the screening plate, giving drug concentrations of 3.3 × 10^3^, 1.1 × 10^3^, 3.7 × 10^2^, 1.2 × 10^2^ ng/mL. The plates were then incubated at 37 °C and 5% CO_2_ for 48 h.

#### 2.5.4. Quantification of Inhibition

Quantitative PCR (qPCR) targeting the 18S rRNA locus was used to assess the in vitro *C. parvum* growth rate. qPCR is widely applied in *Cryptosporidium* infectivity-related research to quantify parasite development [31,32,33,34].

After 48 h incubation, the maintenance medium was aspirated from the wells, and the infected monolayers were washed with 500 µL of sterile PBS (Sigma-Aldrich^®^) twice. Cell monolayers were harvested in 70 µL of 0.5% trypsin/EDTA (Sigma-Aldrich^®^) (pH 8.0). The total genomic DNA (gDNA) was extracted from the infected cells at 48 h post-infection using DNeasy isolation kits (QIAGEN Inc., Valencia, CA) with minor modifications to the manufacturer’s protocol [35]. Briefly, the lysed cells were added to the labelled microcentrifuge tubes. Then, five freeze–thaw cycles were carried out, freezing in liquid nitrogen for 30 s and thawing at 65 °C for 1 min. The final elution volume was adjusted to 50 µL from the manufacturer’s recommended volume of 200 µL for the DNeasy isolation kits.

Quantitative PCR (qPCR) was performed in triplicate for each DNA sample from each plant extract concentration using primers targeting the 18S rRNA locus (5′ AGTGACAAGAAATAACAATACAGG 3′ and 5′ CCTGCTTTAAGCACTCTAATTTTC 3′) [36], and the 6-carboxyfluorescein (FAM)-labelled TaqMan probe (5’ FAM- AAGTCTGGTGCCAGCAGCCGC-BHQ1 3′) [37]. A standard curve was constructed using five triplicates of recombinant plasmids containing partial fragments of the *Cryptosporidium* 18S rRNA, serially diluted at a 1:10 ratio and calibrated by digital droplet PCR (ddPCR) as described by Yang et al. [38].

Reaction conditions were a final volume of 15 µL containing 1x Kapa Taq PCR buffer (KAPA Biosystems), 3.75 mM MgCl_2_, 0.5 µM of each forward (18siF) and reverse (18siR) primers (Fisher Biotec, Australia), 400 µM of each deoxynucleoside triphosphate (dNTP) (Promega, Australia), 0.2 µM Taqman probe, 5 U/µL Kapa Taq DNA polymerase (KAPA Biosystems), and Ultra-Pure PCR grade water (Fisher Biotec, Australia). An aliquot (14 µL) of the template was used in each reaction. Cycling parameters were one pre-melt cycle at 90 °C for 3 min followed by 50 cycles of 94 °C for 20 s and 60 °C for 90 s on a Rotor-Gene Q system (Qiagen, Mortlake, NSW, Australia).

Additionally, an immunofluorescence assay was performed to provide qualitative confirmation of the qPCR results. In this assay, the *C. parvum* infection intensity was qualitatively evaluated using a fluorescein-conjugated specific polyclonal antibody (Sporo-Glo™, Waterborne Inc., New Orleans, LA, USA) according to the manufacturer’s protocol.

#### 2.5.5. DNA Sequencing

A subset of amplicons was sequenced to confirm the *C. parvum* strain used, and that no contamination had occurred. Purified PCR products were sequenced independently in both directions using the secondary PCR primers (Fisher Biotec, Australia), and resultant nucleotide sequences were aligned using the BioEdit v. 7.0.1 package and compared with available DNA sequences of *Cryptosporidium* in the GenBank database using the NCBI BLAST basic local alignment search tool (http://www.ncbi.nlm.nih.gov/BLAST/, accessed on 14 December 2021).

### 2.6. Data Analysis

Data are presented as the mean percentage (±SEM) of three replicates. For cytotoxicity data, all values obtained from cell monolayers treated with plant compounds were normalised to those obtained from the wells containing only fresh media (background control). The per cent cell viability was calculated relative to the untreated control, which has only the HCT-8 cells, using the following formula:Relative viability (%) = [Mean Measurement (treatment) − Mean (background control)/Mean (untreated control) − Mean (background)] × 100

For in vitro growth inhibition data, the percentage of endogenous stages of *C. parvum* in treated groups was compared to untreated control using the following formula:% inhibition = [(DNA copies in positive control well − DNA copies in treated well)/DNA copies in positive control well] × 100

Fifty per cent inhibitory concentration (IC_50_) values, with standard errors, were calculated using the non-linear regression function of GraphPad Prism^®^ version 9.4.0 (673) for Windows, GraphPad Software Inc., La Jolla, CA, USA.

## 3. Results

### 3.1. Cytotoxicity Assay

No apparent cytotoxicity was detected in HCT-8 cells incubated with plant extracts at concentrations ranging from 2 to 500 µg/mL up to 72 h, with all cell viabilities above 88%. At the highest tested dose (500 µg/mL), the extract of *Tribulus terrestris* (caltrop) had the maximum cell viability (99.6% ± 0.4), while the *Cucurbita pepo* (pumpkin) extract had the lowest cell viability (88.1% ± 0.6) (Appendix A). The cell viability data were compatible with the microscopic observations, which showed no cytotoxicity effects of plant extracts on the cell monolayers.

### 3.2. Cryptosporidium Parvum Growth Inhibition Assay

The sequencing results of the randomly selected purified secondary PCR products showed 100% similarity to the *C. parvum* Iowa strain (CP044422) in the NCBI GenBank database.

The average *Ct* values obtained for 20,000 and 2 *C. parvum* oocysts in the standards of the qPCR assays were 21 and 33, respectively. A three-fold increase in *Ct* values was observed among the standards (Figure 1), confirming the reliability of qPCR measurements. Furthermore, our qPCR measurements of parasitic load in the untreated control reflected the success of infection and the parasite proliferation over the experimental period.

#### 3.2.1. Growth Inhibition at a Single Concentration

All plant extracts tested against *C. parvum* demonstrated varying inhibition activities *in vitro*. Based on the qPCR results, the extracts of *C. longa* (turmeric), *P. nigrum* (black pepper), *N. sativa* (black cumin), and *E. ribes* (false black pepper) had the maximum *C. parvum* growth inhibition at 500 ng/mL with 79.6 ± 1.0, 73.6 ± 2.7, 68.1 ± 1.0, and 61.1 ± 1.6, per cent inhibitions, respectively (Table 2). The remaining plant extracts showed growth inhibition values of 50% or less, with *Centella asiatica* (Gotu kola) having the smallest effect on *C. parvum* growth in vitro (Table 2).

#### 3.2.2. Dose–Response Analysis

The extracts of *C. longa*, *P. nigrum*, *E. ribes*, and *N. sativa* exhibited dose-dependent efficacy on *C. parvum* growth inhibition in vitro, with IC_50_ values varying from 3.3 to 141.3 ng/mL. The drug control treatment (trifluralin) had an IC_50_ of 133.9 ng/mL (Figure 2).

The fluorescence-microscope observation revealed the presence of intracellular stages in the infected and treated monolayers and the untreated control wells. Compared to the untreated controls, a higher inhibition of *C. parvum* growth was observed with *C. longa*, *P. nigrum*, *E. ribes*, and *N. sativa* (Appendix A).

## 4. Discussion

Toxicity is a primary concern when testing the clinical efficacy of any new drug candidate [39]. In the present study, all thirteen tested plant extracts had low toxicity profiles, with more than 85% cell viability at the highest dose (500 μg/mL). All these extracts also showed some ability to inhibit the in vitro growth of *C. parvum*, with four (*C. longa*, *P. nigrum*, *E. ribes* and *N. sativa*) having dose-dependent activity similar to or better than the standard drug control, trifluralin.

We emphasise that these results provide only the first step in the search for potential new drug candidates for cryptosporidiosis. The extracts have not yet been fully characterised for their chemical composition. This is an essential next step in determining the mode of action of the extracts and identifying the specific compounds within them which are responsible for the antiparasitic activity.

In the present study, qPCR targeting the 18S rRNA locus [36] was used to assess the multiplication of *C. parvum* in cell culture. The calculation of a standard curve in qPCR is crucial to validate the quality of each experiment and to obtain accurate results. In the current study, a standard curve was constructed using five triplicates of recombinant plasmids containing partial fragments of the *Cryptosporidium* 18S rRNA, serially diluted at a 1:10 ratio and calibrated by digital droplet PCR (ddPCR) [38]. The use of digital droplet PCR enabled an accurate count of the copy numbers of standards and, thereby, the copy number of unknown concentrations (in the samples treated with plant extracts and in the controls).

The aqueous extract of *C. longa* was the most effective plant product in inhibiting *C. parvum* growth in cell culture, with an IC_50_ of 3.2 ng/mL. The effect of *C. longa* crude extracts has not previously been studied on protozoan parasites, although inhibitory activity has been reported against model nematodes and different helminth parasites. Ethanol extract of *C. longa* rhizome had dose-dependent anthelmintic activity against *Haemonchus contortus*, with 78% worm mortality at 2 × 10^8^ ng/mL after 24 h exposure [40] and performed better than the standard anthelmintic drug, piperazine citrate at 1 × 10^7^ ng/mL against Indian earthworm, *Pheretima posthuma* [41]. Methanol extract of *C. longa* rhizome also showed moderate anthelmintic activity against *Caenorhabditis elegans* with IC_50_ of 1.2 × 10^5^ ng/mL [42]. Curcumin, the most abundant phytochemical in the *C. longa* rhizome and a natural polyphenolic compound from *Curcuma* species, was effective against *C. parvum* in vitro with an IC_50_ value of 4.8 × 10^6^ ng/mL [43].

The potent anticryptosporidial effect of *C. longa* in cell culture may be explained by various mechanisms of action. Species of *Cryptosporidium* can generate reactive oxygen species and inflammation as a part of defence mechanisms against host cells [44]. Curcumin has antioxidant and anti-inflammatory properties [45], which may be responsible for the inhibitory activity of *C. longa* on *C. parvum*. Curcumin was also reported to reduce mammalian cellular phospholipase [46,47], and inhibition of phospholipases and arachidonic acid production has previously been shown to reduce infectivity in *Cryptosporidium* spp. [48]. Curcumin is a known histone deacetylation inhibitor [49]. A member of the apicomplexan histone deacetylase family has been identified in *C. parvum* [50]. Histone deacetylase regulates transcription and is considered one of the novel therapeutic targets for antiprotozoal agents [51]. Therefore, it can be argued that *C. longa* extracts (which contain 95% curcuminoids) may inhibit histone deacetylation and thereby alter the proliferation of *C. parvum*.

Crude extracts of *P. nigrum* have not previously been evaluated for their efficacy on GI parasites. However, Chouhan et al. [52] demonstrated significant in vitro inhibitory actions of *P. nigrum* hexane seed extract and ethanolic fractions against the intracellular protozoan parasite *Leishmania donovani*. The activity of *P. nigrum* extracts against *L. donovani* promastigotes is presumed to be mediated through apoptosis, as evidenced by DNA fragmentation and loss of mitochondrial membrane potential [52]. This may be due to a range of bioactive compounds in *Piper* species, such as flavanones, neolignans, dihydrochalcones, chalcones, and alkaloids, which have been claimed to possess antiprotozoal activities [53]. Piperine, an alkaloid, is well known for its potent antioxidant and anti-inflammatory activities [54]. Therefore, piperine has the ability to scavenge free radicals, mainly reactive oxygen species (ROS) [55]. A significant protective role of ROS has been identified in experimental cryptosporidiosis [44]. It seems to be possible that a similar mechanism is involved in the inhibition of growth of the *C. parvum* by piperine in the *P. nigrum* extract.

To our knowledge, extracts of *E. ribes* have not been previously studied for their antiprotozoal activity, although they have previously been reported to show anthelmintic activities. Aqueous and alcoholic extracts of *E. ribes* seed extracts caused a moderate reduction in faecal egg count in sheep experimentally infected (at 6 mL/sheep) with *H. contortus* [56]. An aqueous extract of *E. ribes* fruit, in 3% and 5% concentrations, caused significantly greater in vitro mortality of *P. posthuma* compared to the same concentrations of the anthelmintic drug piperazine citrate [57]. The seed oil of *E. ribes* also caused mortality of *P. posthuma* in vitro at 1 × 10^7^ ng/mL, and the effect was not significantly different from piperazine citrate at the same concentration [58]. Embelin, the main active ingredient of *E. ribes,* showed significantly greater anthelmintic activity against *P. posthuma* at 1 × 10^7^ ng/mL than albendazole at 1.5 × 10^7^ ng/mL [59].

As for *E. ribes*, extracts of *N. sativa* (black cumin) have not previously been tested against protozoan parasites but have been studied for their anthelmintic activities. Al-Shaibani et al. [60] found that aqueous and ethanol extracts of *N. sativa* had ED_50_ values on in vitro egg hatching of a mixture of GI sheep nematodes, including *H. contortus*, of 2.2 × 10^7^ ng/mL and 2 × 10^7^ ng/mL, respectively, and both extracts reduced in vivo egg counts by 200 mg/kg p.o compared to untreated control sheep. In another study, the essential oil of *N. sativa* at 9 × 10^8^ ng/mL significantly reduced the survival of *C. elegans* larval and adult stages compared to negative (1% ethanol) and positive (levamisole at 8 × 10^6^ ng/mL) controls, with larvae being more sensitive than adult worms [61]. The seeds of *N. sativa* contain fixed and volatile oils, rich sources of quinones, unsaturated fatty acids, amino acids and proteins, and traces of alkaloids and terpenoids. Thymoquinone, the active ingredient in volatile oil, has induced apoptosis in cancerous cells through DNA damage [62] and is believed to cause DNA damage to parasite cells through the increased number of free radicals [63]. Another mechanism that inhibits the parasite’s DNA synthesis may be by obstructing the interaction of the histone deacetylase enzyme with chromosomes [64]. Seeds of *N. sativa* also exhibit an immunomodulatory effect by stimulating CD4-positive T-cells and macrophages [65], which may reduce inflammatory damage.

## 5. Conclusions

Our results demonstrate that the crude extracts of *C. longa*, *P. nigrum*, *E. ribes* and *N. sativa* inhibited *C. parvum* intracellular growth with promising IC_50_ values. These plants offer a great hope to obtain anti-cryptosporidial compounds that are more promising and more effective than current treatments. Based on the sensitivity of parasites to these plant extracts in cell culture, it can be argued that the target sites could be developmental stages. The findings of the cytotoxicity assay showed that the host cells are unlikely to be targeted. However, it is possible that the extracts could target the infected host cells in preference to uninfected host cells. However, these encouraging findings are limited to the in vitro model of *C. parvum* infection. Further studies with appropriate in vivo models of cryptosporidiosis are required to confirm whether these plants could be developed into new anti-cryptosporidial drugs. Moreover, the exact mechanisms of *C. parvum* inhibitory actions of these plants remain to be explained. Future studies to characterise the mode of action of these compounds could lead to the identification of novel *Cryptosporidium* drug targets.

## Figures and Tables

**Figure 1 pathogens-12-00061-f001:**
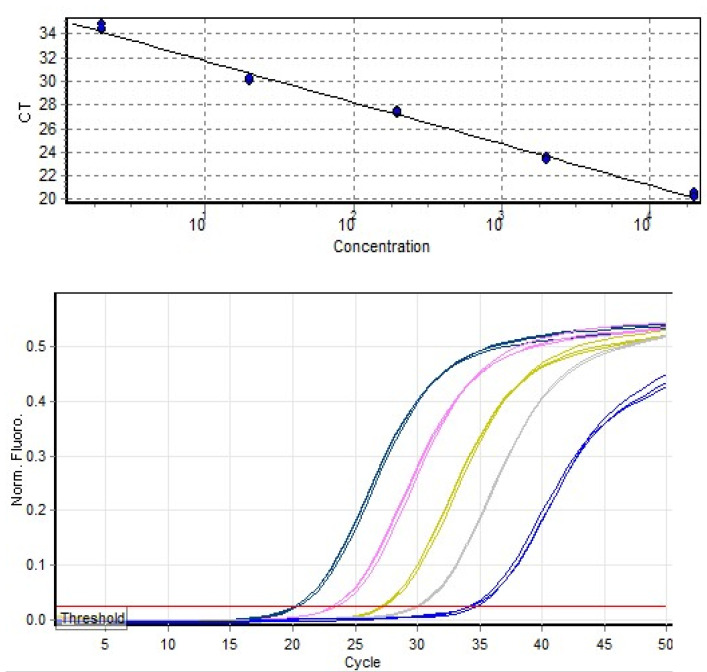
Standard curve and quantitation data generated from serially diluted *Cryptosporidium parvum* DNA standards using RotorGene 2000 software Version 4.7, R^2^ = 0.99.

**Figure 2 pathogens-12-00061-f002:**
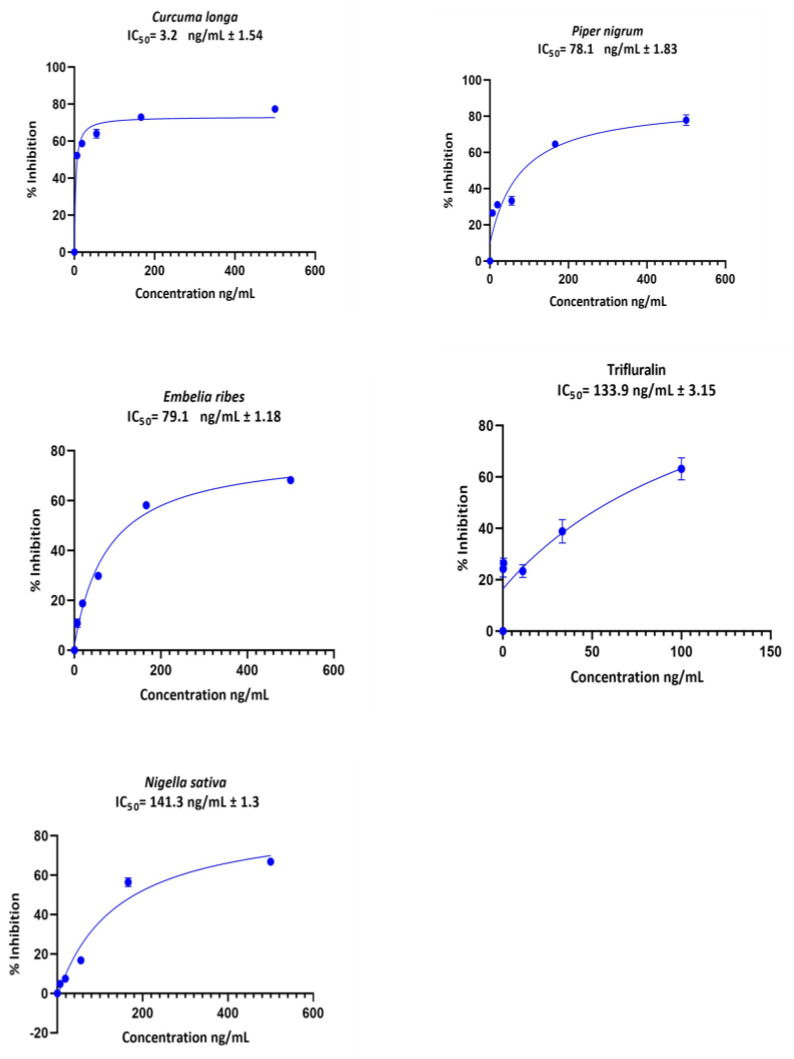
*Cryptosporidium parvum* growth inhibition curves derived from non-linear regression in GraphPad Prism^®^ software for the extracts of *C. longa*, *P. nigrum*, *E. ribes*, and *N. sativa and the control drug Trifluralin*. The bar represents the standard error of means from three replicates.

**Table 1 pathogens-12-00061-t001:** Details of the thirteen plant extracts used in the study.

Plant Species (Common Name)	Family	Part Used	The Solvent Used for Extraction	Characterised Main Phytochemicals	Amount (%*w*/*w*)
*Allium sativum* L. (garlic)	Amaryllidaceae	Bulb/Cloves	70% Methanol	Alliin	1.1
				Allicin	0.55
				Volatile oil	1.2
*Boswellia serrata* Roxb. (Indian frankincense)	Burseraceae	Oleo-gum resin	70% Ethanol	Acetyl keto beta boswellic acid	30.72
				Beta boswellia acid	5.40
				Acetyl beta boswellic acid	2.96
				Acetyl beta boswellic acid	1.57
*Centella asiatica* (L.) Urban (Gotu kola)	Apiaceae	Leaf	70% Ethanol	Total Terpenes (sum of Asiaticoside, Madecassoside, Asiatic acid, Madecassic acid)	46.20
*Curcuma longa* L. (turmeric)	Zingiberaceae	Rhizome	Water	Total Curcuminoids	95.16
*Cucurbita pepo* L. (pumpkin)	Cucurbitaceae	Seed	Water	Glycosides	5.33
*Embelia ribes* Burm. f. (false black pepper)	Primulaceae	Fruit	70% Methanol	Tannins	5.20
*Glycyrrhiza glabra* L. (liquorice)	Fabaceae	Root	80% Ethanol	Glycyrrhizinic acid	50.82
*Moringa oleifera* Lam. (drumstick)	Moringaceae	Leaf	Water	Total Protein Content	22.08
				Saponins	10.2%
				Alkaloids	0.22%
*Nigella sativa* L. (black cumin)	Ranunculaceae	Seed	Water	Total Saponins	12.98
				Total Bitters	4.00
*Piper nigrum* L. (black pepper)	Piperaceae	Fruit	Acetone/hexane	Piperine	95.67
*Thymus vulgaris* L. (thyme)	Lamiaceae	Seed	Water	Total Volatile Organic Compounds	3.23%
*Tribulus terrestris* L. (caltrop)	Zygophyllaceae	Entire plant	Water	Total Saponin Content	41.43
*Vitex negundo* L. (Chinese chaste tree)	Lamiaceae	Leaf	Water	Total Glycoside	26.98

**Table 2 pathogens-12-00061-t002:** Effects of the plant extracts on *C. parvum* oocyst growth inhibition at a single concentration (500 ng/mL) at the end of the 48 h incubation period. Per cent growth inhibition is compared to the negative control (HCT-8 in RPMI and 0.5% DMSO).

Compound	% Inhibition ± SEM
*Curcuma longa* L. (turmeric)	79.6 ± 1.0
*Piper nigrum* L. (black pepper)	73.6 ± 2.7
*Nigella Sativa* L. (black cumin)	68.1 ± 1.0
*Embelia ribes* Burm. f. (false black pepper)	61.1 ± 1.6
*Allium sativum* L. (garlic)	50.3 ± 0.1
*Tribulus terrestris* L. (goat’s-head/caltrop)	50.3 ± 0.5
*Thymus vulgaris* L. (thyme)	50.2 ± 2.6
*Moringa oleifera* Lam. (drumstick)	46.3 ± 2.4
*Boswellia serrata* Roxb. (Indian frankincense)	39.8 ± 0.9
*Glycyrrhiza glabra* L. (liquorice)	38.7 ± 0.7
*Vitex negundo* L. (Chinese chaste tree)	35.2 ± 2.9
*Cucurbita pepo* L. (pumpkin)	31.9 ± 0.4
*Centella asiatica* (L.) Urban (Gotu kola)	22.5 ± 0.2

## Data Availability

Not applicable.

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
