# Peer review of "In Vitro Susceptibility of Cryptosporidium parvum to Plant Antiparasitic Compounds"

_pathogens, 2022, doi:10.3390/pathogens12010061_

Round 1

Reviewer 1 Report

Comments: The objective of current study was to test the plant extracts for their ability to inhibit the growth of C. parvum, established in vitro on HCT-8 cells. The extracts were chosen because of their commercial availability, diversified pharmacological properties, and traditional use against various diseases, including parasitic gastrointestinal (GI) disorders. The study is attractive with good value in scientific field. However; below mentioned points should be improved

Line 47-48: The search for natural antiparasitic products is widely based on traditional medicinal plants used to treat infections “modify  to”  Traditional medicinal plants and their derivatives have showed favorable results against various parasitic infection widely “and add” recent references such as https://doi.org/10.3390/life12030449; https://doi.org/10.12681/jhvms.28497 and  Pak Vet J, 37(2): 145-149.

Line 54-55: C. parvum, with human ileocaecal adenocarcinoma (HCT-8) cells proven to be a suitable cell line “modify to” C. parvum, with human ileocaecal adenocarcinoma (HCT-8) cells have proven to be a suitable cell line

Line 58-59: The extracts were chosen because of their commercial availability, diversified pharmacological properties “add” comma after properties

Line 161-162: Rephrase the sentence: Trifluralin has been shown to be effective in inhibiting the growth of C. parvum in cell culture with an IC50 of 650.4 mg/L

Line 204-206: Rephrase the sentence: To confirm the C. parvum oocysts strain used and that no contamination had occurred, a subset of amplicons was sequenced

Line 257: Per cent growth inhibition is presented compared with the negative control (HCT-8 in RPMI and 0.5% DMSO) “modify to Per cent growth inhibition is compared to the  negative control (HCT-8 in RPMI and 0.5% DMSO).

Line 271: In the present study, all thirteen tested plant extracts had low toxicity profiles “add” comma after profiles

Line 368-476: Check the references carefully following the journal format

Comment: How can compare this study to others in novelty point of view? Many studies have done using medicinal plants against various infectious diseases including parasitic infection?

Reviewer 2 Report

The present study provides information in relation to testing the efficacy of some natural plants (n=13) against HCT-8 with Cryptosporidium parvum in vitro. The major serious flow of the present work is the chemical characterization of the plant materials, without characterization of the used substances, data cannot be of high significance, particularly that some previous reports revealed the anti-cryptosporidium activity of some of the studied plants already (https://pubmed.ncbi.nlm.nih.gov/23347204/ & https://link.springer.com/article/10.1007/s00436-009-1535-5) . Recommendation of use some unknown or uncharacterized plants for treatment of a diseases can not be performed without presence of a clear characterization.

Hereby my major comments:

1.    Novelty of the presented work should be included in the abstract.

2.    The Introduction section should be extended since it is very short in the current form and please consider including more data about the zoonotic relevance of the studied parasites and the epidemiological profile of the disease which are important information related to the knowledge of the subject of experimental work.

3.    Research question and its gap should be added in the end of the introduction section.

4.    The scope of experimental work must also be extended to include testing /standardization of the composition of the parts used of the extracts used of the applied plants. Without this step, this paper Should Not Be accepted.

5.    Explaining the potential mechanisms of the noted effect of their anti-cryptosporidium activity.

6.    Address of the source of parasite material (BioPoint Pty. Ltd) and American Type Culture Collection should be added.

7.    Lines 88-90 has an explanation, they should be moved to discussion section.

8.    On what base you did unify the maximum concentration used (500 µg/mL) for this numerous number of plants? Logically concentration and effects should be different.

9.    Molecular data should be explained in the discussion section.

10. Where is your study limitations? No study without limitations and please consider addition of this section.

11. Please consider deletion of  lines 338-340 in conclusion ‘’These plants 338 may therefore be attractive anti-cryptosporidial agents, particularly for developing countries which may not be able to afford new-generation drug’’. Your findings are just in vitro results. This section should be rewritten.

12. The Word files in your supplementary material unreadable for reviewer. They should be fixed.

13. Extensive editing of English language and style required since manuscript has dozens of linguistic errors.

Reviewer 3 Report

The manuscript entitled "In vitro susceptibility of Cryptosporidium parvum to plant antiparasitic compounds" reports a study of the extracts from 13 different plant species that provide evidence for inhibiting the growth of Cryptosporidium parvum in vitro. Additionally, the cytotoxic effect of extracts on HCT-19 8 cells was evaluated.  Four plant species with the strongest evidence of activity against C. parvum.

There are a few points to clarify/change.

1 - Materials and Methods

"Traditional Indian Medicine (Ayurveda)"  Please, add some references regarding the use of these plants (and their part) in antiparasitic treatments.

2 - Figure 1 -

Please, improve significantly the caption of figure 1. 

3 - Figure 2

Please, improve the plots in figure 2.

4 - Perhaps, some comparative multivariate analysis of some NMR spectra, mass spectra, or even the chemical composition of the extracts could show some synergism and the main secondary metabolites responsible for the activity against C. parvum.

Reviewer 4 Report

The authors evaluated the anti-cryptosporidium ability of 13 plant extracts in vitro by quantitative PCR and immunofluorescence assays. In addition, the author found that the four plants with the strongest resistance to C. parvum are Curcuma longa, Piper nigrum, Embelia ribes, and Nigella sativa, all with dose-dependent efficacy. However, the manuscript needs major revisions. The immunofluorescence pictures are also a bit blurry; I would suggest to take clearer pictures which will make the manuscript look more professional.

1. The plant extract Allium sativum L. (garlic) in Table 1 is shown in bold. Is there any special meaning? If not, please unify the format.

2. Line 111-112, Plant extracts were dissolved in DMSO, therefore, DMSO should be used as negative control.

3. In figure S1, fluorescent photos are very unclear. In addition, Clear pictures showing bright field images and DAPI nuclei staining should be combined. Pictures with better resolution need to be provided.

4. To my knowledge, the common control drugs at present are paromomycin or nitazoxanide.

5. Did the author wash out the uninfected oocysts and sporozoites after adding cryptosporidium oocysts and incubating for 48 hours? How to clean it? How many times? Please specify, because this is crucial to qPCR results.

6. Line 199-202, the author uses an immunofluorescence assay to confirm the results of qPCR. Is it to measure one field or multiple fields for statistical analysis? Please be more elaborate on the statistics section. In addition, the result of immunofluorescence assay is not satisfying; this part needs to be revised.

7. Line 232,It looks like Supplementary material 2 here should be Supplementary material 3.

8. Line 256, What is the single concentration in the header of Table 2? Or display in the table

9. Line 266-269, please show the experimental results of this part. How was the statistical analysis performed? According to the context, the Supplementary material 3 here should be Supplementary material 2, please revise.

10. Line 341-342, from this study, it can not be determined that the target of the drug is the development stage of the cryptosporidium, which may also be the host cell itself.

Round 2

Reviewer 1 Report

Line 56-59: Traditional medicinal plants and their derivatives have shown favourable results against various parasitic infections "here" add at least two to three references as recommended in previous comments.

Line57-59: In vitro screening is a critical step in the early stage of drug development. Compared to animal models, in vitro models are less expensive, less time-consuming, and more convenient for screening drugs for their efficacy against parasites "here" add recent references 

Reviewer 2 Report

The authors worked on improvement of the manuscript but they failed to address my  major concern in relation to the chemical composition of the used p ant material. Without chemical characterization, paper can not be accepted. In science, we cannot use a certain plan materials or extract without chemical characterization. Their reply was ‘’ All the plant extracts were standardized and validated by the manufacturers using High-performance liquid chromatography (HPLC).  However, they did not include any chemical characterization of the plant materials. Without this important issue, manuscript cannot be accepted. Furthermore, I asked the authors for their own suggested mechanisms for their presented work, not those available on literature and they did not address this point. Therefore, my suggestion is to reject the manuscript and encourage its resubmission after chemical characterization of the used plant materials.

Reviewer 4 Report

The authors already answered my questions, and revised them in the text.

Author Response

Thank you.

Round 3

Reviewer 2 Report

The authors failed to address my major two concerns during the revision:

1- Firstly, in relation to the chemical composition of the used plant material. In their response, they wrote '' this was beyond the scope of the current study. Our aim, as is clear from the Introduction, was to screen a large number of extracts for potential antiparasite activity, given the current lack of efficacious drug treatments for Cryptosporidium parvum. Having identified four extracts with very promising activity against C. parvum''.

However, I am still convinced that without chemical characterization, paper cannot be accepted. As I mentioned in my previous comments, we cannot use some plan materials or extracts WITHOUT obvious chemical characterization. Otherwise, there will be thousands of papers on unidentified plant extracts.

2- Authors also failed to address my comment in relation to the suggested mechanisms underlying the action of the most effective four extracts, at least, and they just focused on mentioning the available information in the literature. They also included incomplete Table 1 in the manuscript.

Without this important issue related to chemical characterization, manuscript cannot be accepted. Therefore, my suggestion is to reject the manuscript and encourage its resubmission after providing a clear chemical characterization of the used plant materials.